# Use of Flavin-Related Cellular Autofluorescence to Monitor Processes in Microbial Biotechnology

**DOI:** 10.3390/microorganisms10061179

**Published:** 2022-06-08

**Authors:** Lucie Müllerová, Kateřina Marková, Stanislav Obruča, Filip Mravec

**Affiliations:** Faculty of Chemistry, Brno University of Technology, Purkynova 118, 612 00 Brno, Czech Republic; katerina.markova@vut.cz (K.M.); obruca@fch.vut.cz (S.O.); mravec@fch.vut.cz (F.M.)

**Keywords:** bacteria, green autofluorescence, flavins, viability, average fluorescence lifetimes

## Abstract

Cellular autofluorescence is usually considered to be a negative phenomenon because it can affect the sensitivity of fluorescence microscopic or flow cytometric assays by interfering with the signal of various fluorescent probes. Nevertheless, in our work, we adopted a different approach, and green autofluorescence induced by flavins was used as a tool to monitor fermentation employing the bacterium *Cupriavidus necator*. The autofluorescence was used to distinguish microbial cells from abiotic particles in flow cytometry assays, and it was also used for the determination of viability or metabolic characteristics of the microbial cells. The analyses using two complementary techniques, namely fluorescence microscopy and flow cytometry, are simple and do not require labor sample preparation. Flavins and their autofluorescence can also be used in a combination with other fluorophores when the need for multi-parametrical analyses arises, but it is wise to use dyes that do not emit a green light in order to not interfere with flavins’ emission band (500–550 nm).

## 1. Introduction

Single-cell analysis pervades many modern scientific branches such as genomics, proteomics, transcriptomics, epigenomics, and metabolomics. The methods used are also various—from DNA sequencing of one cell (single-cell DNA sequencing or also scDNAseq) in genomics, a combination of antigens and quantum dots in proteomics, to spectroscopic methods in metabolomics. The last of the listed also includes various methods and techniques based on fluorescence detection. This work used two of these techniques—fluorescence microscopy and flow cytometry [1].

Avoiding the introduction of exogenous probes means that the cellular environment can be maintained. The fluorescence exhibited by these endogenous fluorophores is called autofluorescence. The most important autofluorescent chromophores are nicotinamide adenine dinucleotide (NADH); flavin adenine dinucleotide (FAD); aromatic amino acids such as tryptophan; lipofuscins; advanced glycation end products; and some proteins, e.g., collagen [2]. Every such chromophore has its excitation and emission spectra that are characteristic of that particular molecule, and so it is quite easy to distinguish between them.

It is very important to have access to analyses that are fast, reliable, and easy, with minimal changes to the culture of the interest. Staining the cells is, of course, reliable, and relatively fast, but having the option to use autofluorescence, i.e., the native ability of certain molecules in the cell to emit light, is a path that saves time, with minimal intervention to the cells’ natural processes.

Generally, autofluorescence is considered to negatively affect flow cytometric or fluorescence microscopic analyses because of the wide spectra emission. However, many teams have used autofluorescence as a powerful tool in their experiments. For example, Lingling Yang’s team used autofluorescence for the detection and quantification of bacteria [3], Surre et al. proved that an increase in bacterial autofluorescence mirrors the cells’ struggle for survival under various stress factors [4], and there are even studies that employ red channel autofluorescence to analyze cell aging [5]. Autofluorescence has also been used as a marker of viability, for example, in *Bacillus anthracis* spores aerosol [6], *Listeria monocytogenes*, *Salmonella enteritidis*, and *Escherichia coli* [7], using red and green autofluorescence for identifying viable cells of *Synechocisis* sp. PCC 6830 [8].

The use of fluorescence arising from endogenous chromophores in cells and tissues has received much attention in recent years. For instance, a great work that used deep-UV excitation (200–250 nm) was used to detect and differentiate between bacteria on various opaque surfaces, where the molecules emitting the fluorescence were a mix of intrinsic proteins, amino acids, nucleic acids, flavins and other aromatic compounds that were concentrated in the cell [9]. Using FLIM (Fluorescence Lifetime Imaging Microscopy) and detecting ultraviolet-induced (UV-induced) autofluorescence of reduced coenzymes such as NADH and NAD(P)H can be used to get a metabolic fingerprint of various bacteria, such as *Escherichia coli*, *Bacillus subtilis*, *Pseudomonas aeruginosa*, or *Salmonella enterica*, as was recently demonstrated by Bhattacharjee and Datta. In their study, the team mapped the metabolic activity of the bacteria mentioned within natural communities and medical infections in order to provide insights into the role of bacterial metabolism and whether it can determine bacterial behavior in cultures and communities. They concluded that the ratio of free to bound NAD(P)H that correlated to the ratio of longer and shorter fluorescence lifetimes depends on species, growth phase, and history of exposure to nutrient media and antibiotics [10]. Furthermore, Surre et al. found that autofluorescence can be used as a marker of the specific stress response because, when *E. coli* was exposed to various stress situations, an increase in expression of genes encoding for diverse autofluorescence-inducing flavoproteins was observed [4].

Analysis of autofluorescence is a non-destructive, non-contact, and quite fast technique for the identification of cells. On the other hand, the resulting signal is usually weak; sometimes it can be influenced by the emission of the other substances present in a sample [11,12].

*Cupriavidus necator* H16 is a Gram-negative facultative chemolithotroph bacterium that belongs to the ß-proteobacteria class. Because of its ability to synthesize polyhydroxyalcanoates (PHAs), it is one of the model organisms for PHAs biosynthesis and intracellular degradation. Its mutant strain, *Cupriavidus necator* PHB-4, does not have the ability to produce PHAs [13].

PHAs are organic polyesters that are synthesized by the bacteria when there is an excess of carbon substrate in the environment. PHAs are stored intracellularly in the form of granules. These polyesters have a hydrophobic character which is important because the fixation of PHAs does not increase osmotic pressure in the cytoplasm [14]. These polymers resemble some petrochemical polymers in their properties, but in contrast to them, PHAs are fully biodegradable, as their primary use in bacterial cells is a storage of energy and carbon. PHAs such as poly(3-hydroxybutyrate) (PHB) and poly(3-hydroxybutyrate-co-3-hydroxyvalerate) (PHBV) show biodegradable characteristics in numerous environments. Generally, PHAs are considered to be renewable and sustainable alternatives to petrochemical polymers; nevertheless, the major disadvantage of PHAs is their high cost of production, which is associated with the demanding process of fermentation. Therefore, fast and reliable analytical tools that enable us to control and monitor the fermentation process can contribute to decreasing the cost of the final product [15].

Quick and reliable detection of PHAs is important for finding new techniques for the cultivation of bacteria with maximal PHAs content and yields, not only when searching for novel PHA-producing microorganisms, but also when optimizing and improving existing biotechnological processes. Generally, PHAs production is a very dynamic process and is directly related to numerous external factors [16]. The usual method employed when monitoring PHAs production is flow cytometry and the use of fluorescent stains; for example, Li and Wilkins used Nile Red and developed a very useful model for the quantification of PHA production in time [17]. Similarly, Saranya et al. also used Nile Red staining and developed a protocol for quantification of PHA production during the whole bioprocess [18]. Contrarily, Karmann et al. used a different staining strategy and employed dual staining with BODIPY 493/503 and SYTO 62 for PHA quantification, without the need of washing the cells after cultivation [19].

Flavins, which include riboflavin and its derivates, consist of the isoalloxazine ring system where a ribityl side chain is attached to the central *N*-10 position in the pyrazine moiety; flavin mononucleotide (FMN) and flavin-adenine nucleotide (FAD) are involved in various redox reactions. For example, oxidized FAD plays a key role in the conversion of energy from acetyl-CoA to ATP (adenosine triphosphate). Pure flavin-free derivates, FMN and FAD, exhibit a very strong fluorescence in aqueous solution at pH of about 7, with excitation/emission maxima of about 440–450 nm and 525 nm respectively, according to the literature. [16,20,21]

Mihalcescu et al. have shown that autofluorescence at a wavelength 525 nm in *E. coli* is exhibited by approximately 80% FAD, FMN, and riboflavin; the other 20% is flavin derivates [22].

In this work, we focused on endogenous chromophores of the flavin variety, mainly because of the possible link between the role of flavins in cell metabolism and PHA metabolism. There is a strong suggestion to use autofluorescence as a marker in biotechnological processes. Here we show proof that “green” autofluorescence in *C. necator* is caused mainly by flavins, and it is possible to use this chromophore as a marker of the physiological state of the microbial culture; furthermore, an analysis of autofluorescence can also be utilized in combination with other green fluorophores when paying attention to the average fluorescence lifetimes of the particular dyes.

## 2. Materials and Methods

### 2.1. Microorganisms and Cultivation

The wild type of *Cupriavidus necator* H16 (CCM 3726) was purchased from the Czech collection of microorganisms, Brno, Czech Republic. PHA non-producing mutant *Cupriavidus necator* PHB-4 (DSM-541) was purchased from Liebnitz Institute DSMZ-German collection of microorganisms and cell cultures, Braunschweig, Germany. After growing on agar dishes (Nutrient Broth media—10 g beef extract, 10 g peptone, and 5 g NaCl, 20 g agar in 1 L of distilled water), *C. necator* cells were grown in the media in Erlenmeyer flasks (50 mL of media in 250 mL flasks) for 24 h, shaken at 140 rpm (Nutrient Broth medium—10 g peptone, 10 g beef extract, and 5 g NaCl in one liter of distilled water; total concentration was 25 g·L^−1^). Then they were inoculated into standard production media for 72 h (Erlenmeyer flasks, 100 mL media in 500 mL flasks) ((NH_4_)_2_SO_4_, 3 g; KH_2_PO_4_, 1.02 g; Na_2_HPO_4_, 11.10 g; MgSO_4_, 0.20 g; fructose, 20 g; and trace elements solution, 1 mL, in 1 L of distilled water) (the trace element solution was composed of 9.7 g FeCl_3_, 7.8 g CaCl_2_, 0.156 g CuSO_4_·5 H2O, 0.119 g CoCl_2_, and 0.118 g NiCl_2_ in 1 L of 0.1 M HCl), at a temperature of 30 °C.

### 2.2. Sample Tubes Preparation for Fluorescence Analysis

For the cell samples, 1 mL of the culture sample was pipetted into 2 mL tubes and washed with PBS two times (centrifugation 8000× *g* for 5 min, lab temperature); the cell count was determined by flow cytometry (see Section 2.5, below, for the instrument specifications), and the samples were then diluted with PBS to get the appropriate concentration in 1 mL. A suspension with 10^6^ cells in PBS was prepared.

Viability experiments included dead cells; the cells were put into the water, at 95 °C, for 15 min. Control tubes contained only PBS, at a pH of 7.4.

Flavin standards’ concentration was 10^−6^ mmol in sterile DI water, and they were stored in the dark at 4 °C. All flavin standards were purchased from Sigma-Aldrich/Merck (Germany): riboflavin—CAS, 83-88-5; FAD—CAS, 84366-81-4; and FMN–CAS, 130-40-5.

### 2.3. Propidium Iodide Staining Essay for Flow Cytometry

We used PI staining for experiments for flow cytometry. The concentration of the stock solution was 1 mg·mL^−3^ in distilled water. The solution was stored in the dark at a temperature of 4 °C. To 1 mL of cell suspension, 5 µL of PI was added. After 15 min of incubation in the dark, the sample was centrifuged (8000× *g*, 5 min, lab temperature) and then washed with PBS buffer two times; 1 mL of PBS buffer was then added to the cells to resuspend the pellet. The laser used was 488 nm, and the signal was detected in the red channel, 690 ± 35 nm

### 2.4. BODIPY 493/503 Staining Essay

The green BODIPY dye was purchased from Thermo Fischer Scientific (Massachusetts, USA) CAS: 121207-31-6. Stock solution concentration was 1 mg·mL^−3^; the solvent was anhydrous DSMO, from which 10 µL was added to 1 mL of cell suspension. The incubation time was 30 min; the tube was placed in a dark place, at lab temperature. After that, the cells were rinsed with 1 mL PBS buffer two times (centrifugation 8000× *g* for 5 min, room temperature), and then 1 mL of PBS buffer was added in which the pellet was resuspended. This procedure was followed when preparing samples for both flow cytometry and fluorescent microscopy.

When executing flow cytometry experiments, the samples were illuminated by a 488 nm laser; fluorescence was then captured in the same channel as autofluorescence, i.e., 535 ± 35 nm.

### 2.5. Flow Cytometry

The flow cytometer used was Apogee A50 (Apogee Flow Systems, Great Britain). The concentration of cells was 10^6^ in 1 mL for every sample measured (see Section 2.2 for details on the sample preparation). A 488 nm laser was used; autofluorescence was collected in the green channel of the cytometer, 535 ± 35 nm. When using BODIPY 493/503, the collecting channel and laser were the same as in the case of collecting autofluorescence. When using PI, we also collected fluorescence in the red channel 690 ± 35 nm, with the laser also being 488 nm.

The sampled volume was 100 µL, and drawing speed was 19 µL·s^−1^. First, we used cytogram SALS (peak)/SALS (area) to localize the cell populations; the collecting channel was then selected accordingly to the sample—a histogram that was either a sum of the height of fluorescence intensity peaks or the area under said peaks. The PMT value that was established for every analysis was 250; however, it was a little tricky, considering that we had to detect autofluorescence and the green fluorescence of the BODIPY 493/503 dye, whose fluorescence intensity is much higher, with the same PMT value.

### 2.6. Fluorescence Microscopy, FLIM

The fluorescence microscope used was MicroTime 200 (PicoQuant GmbH, Germany) and the spectrofluorometer was Fluorolog (HORIBA Scientific, Pennsylvania, USA). The sample was diluted to a concentration of 3% cells in PBS buffer, at pH 7.4. We used a 467 nm laser, and a 520 ± 17.5 nm emission filter and 470/635 dichroic mirror for detection were chosen. The frequency of the laser was 40 MHz, which made the laser pulse 25 ns. Water immersion ultraplanar superapochromatic UPLanSApo 60×/W objective was chosen. Detection canals τ-SPAD with noise under 100 Cnts∙s^−1^ were used. The sample preparation is described in Section 2.2. Sample tubes were prepared for fluorescence analysis.

## 3. Results

### 3.1. Analysis of Emission Spectra of Flavins and Whole Bacterial Cells

Flavins’ fluorescence maxima according to the literature are as follows: for FMN, E_ab_ = 370,450 nm and E_em_ = 495,520 nm; and for FAD, it is E_ab_ = 450 nm and E_em_ = 510 nm [23,24,25].

First, we measured the spectra of the flavins’ standards prepared to possibly differentiate between their emission spectra when using the spectrofluorometers and microscope (see Figure 1). The idea was to use FMN’s, FAD’s, and riboflavin’s single emission spectra, compare these data with the cells’ spectra when under different conditions (different viability for this study, and also useful for future stress experiments), and see which flavin’s intensity differs in time under different conditions.

The peak was wide and found to be localized in the region of 510–550 nm, which correlates with the literature. Unfortunately, the three flavins’ spectra were almost identical, meaning that we could not use their spectra to differentiate between the flavins or compare them; one has to consider the autofluorescence of all three as one singular value. We encountered the same issue when trying to differentiate between these molecules based on their average fluorescence lifetimes, where we also could not determine any distinction between the flavins. Galbán et al. [26] reported the average fluorescence lifetime of riboflavin to be 5.06 ns, FMN 4.7 to be ns, and FAD to be 2.27 ns. This is supported by Albani, who stated that τ_FAD_ = 2.3 ns and τ_FMN_ = 4.7 ns [27]. Although we did not get the same discernment in this particular experiment, we do strongly agree with these values because of the average fluorescence lifetimes we concluded to be connected with flavins in the cells (see Section 3.3).

Using the same approach, we analyzed the autofluorescence emission spectra of the bacterial cells of *Cupriavidus necator* H16 and its PHAs non-accumulating mutant strain *Cupriavidus necator* PHB-4; the results are demonstrated in Figure 2.

The fluorescence maximum is found at approx. 515 nm. These spectra resemble that of flavins, and, therefore, flavins are considered to be the predominant source of the cells’ green autofluorescence.

### 3.2. Using Green Autofluorescence to Monitor the Viability of Prokaryotic Cell Cultures

In this experiment, four samples were investigated—viable and non-viable cells (see Methods) of the two bacterial strains, PHB producing wild-type *C. necator* H16 and its non-PHB producing mutant, *C. necator* PHB-4. The intensity of green autofluorescence was measured.

At first, we measured the excitation spectra for an emission of 525 nm, searching for other possible excitation wavelengths which could be applied. The reason for this was the possible use of other excitation laser wavelengths which could result in more possibilities when it comes to combination analyses. Two excitation maxima were identified—one in the range of 360–370 nm and another around 440 nm, as can be seen Figure 3. These two peaks were also found by Galban et al. [26] when analyzing FAD spectra and also by Schmidt when researching the bacteria *Legionella rubrilucens* and riboflavin’s excitation spectra [28].

We then used an excitation wavelength in the UV range of 266 nm to see if the autofluorescence of *C. necator* H16 and *C. necator* PHB-4 change, depending on their viability status, seeing as UV excitation under 300 can be used to excite flavin molecules [29]. Le-Pan also used a 266 nm laser to obtain emission spectra of riboflavin and other organic compounds such as NADH [30]. As demonstrated in Figure 4, the intensity was the same in all four cases—around 55 µJ. The emission maxima were in the interval from 500 to 550 nm. There cannot be seen any shift dependent on the cells’ viability, with the exception of *C. necator* H16 dead cells sample, in which the emission maximum was shifted from the original live culture by about 40 nm.

The idea behind this experiment was to use the UV excitation for viability experiments to see if it is even possible with the investigated bacteria to get a trend or a simple fingerprint that could be used in biotechnological processes, or if we are limited only to the visible-light wavelengths. As can be seen in Figure 4, the UV excitation for viability experiments cannot be applied, because the differences between the samples are just too insignificant. Nevertheless, autofluorescence can be used as a marker of the presence of the cells when combined with fluorescence dyes that are excited in the UV range. This might be very useful, for instance, when one needs to distinguish cells from abiotic particles during flow cytometry analysis of the complicated microbial samples in which the presence of solid microparticles can be expected.

UV excitation can be used in certain cases and experiments when using UV excited stains such as LIVE/DEAD Fixable blue dead cell stain kit by Thermo Fischer or Hoechst 33342; however, we could not use it in our case as a lone method for defining bacterial viability. As a result, we explored the visible wavelengths lasers area possibilities only.

As can be seen in Figure 5, the fluorescence of live cells is remarkably brighter (up to ten times the photon count). The answer to the question of why the dead cells still emit autofluorescence can be because there are certain metabolic pathways that are still partially active, or autofluorescent molecules could still be present for some time, seeing as, after 24 h, the non-viable cells were not emitting any kind of fluorescence (data not shown).

When using flow cytometry, we stained the cells with propidium iodide to confirm the information related to viability measurement obtained by green autofluorescence; the results are shown in Figure 6. Our objective here was to use autofluorescence combined with a fluorescence dye that has an emission maximum relatively further away from the cellular autofluorescence maximum. Propidium iodide is widely used when gaining data correlated to the cells’ viability, so this experiment could indicate whether the cells’ viability is also reflected by the cellular autofluorescence intensity. Column (A) demonstrates green cellular autofluorescence, and Column (C) shows propidium iodide fluorescence. This experiment proved that green cellular autofluorescence is lower in *C. necator* PHB-4 cells than in *C. necator* H16 live ones, and the photon count for both dead cells samples was lower compared to the live cells.

### 3.3. Analysis of Average Fluorescence Lifetimes

Time-correlated single-photon counting (TCSPC) is a statistical method that is based on the summation of individual photons collected by detectors from the moment the signal starts the light impulse, with high frequency emitting enough photons in this case. Most commonly, a laser is used. The result of this method is the extinction curve of a fluorophore, also called average fluorescence lifetime. This is a way how the FLIM method gets information about various fluorescence lifetime distributions, and because of TCSPC, it is possible to map cells based on different fluorescence lifetimes of a fluorophore [31].

During our experiments, a very interesting phenomenon occurred. While mapping the cells, a trend had been observed—the cells, even though they were unstained, always demonstrated certain average fluorescence lifetimes. It was interesting mainly because the average fluorescence times did not change, and they were present in viable and also non-viable cells. We have also established how exactly it is possible to use them in combination with average fluorescence lifetimes of other fluorescent molecules—fluorescent dyes, for example—in experiments described below.

We identified three different average fluorescence lifetimes: 3.8–4.2 ns, 0.8–1.2 ns, and one that was several picoseconds long, as can be seen in Figure 7 and Figure 8. However, because the last lifetime mentioned was too short, we focused mainly on the two longer lifetimes and further investigated them. The shortest time is not a problem for the experiments per se, but since it is most probably heavily influenced by physical phenomena such as scattering and flavins’ rotating—or it could be the fluorescence lifetime of the stacked conformation of FAD, as was mentioned before—we focused on the two average fluorescence lifetimes that were in the ranges of 3.8–4.2 ns and 0.8–1.2 ns [32,33,34].

Interestingly, in all *C. necator* H16 samples, we could clearly observe outlines of PHB granules inside the cell. The average fluorescence lifetime imaging revealed that the outlines had the highest values—4.800 ± 0.015 ns. As was written before, the average fluorescence lifetime could belong to the open conformation of FAD. The fact that no other average fluorescence lifetimes are concentrated in the area of outlines of PHA granules is very interesting and could be further investigated in follow-up experiments.

Figure 9*C. necator* H16 and Figure 10
*C. necator* PHB-4 show cells non-stained (left) and stained with BODIPY 493/503 (right). Expectedly, the intensity of fluorescence of the stained samples is higher than that of non-stained, but what is interesting is the fact that the intensity values of the shorter average lifetime, τ_1_, rose only slightly.

## 4. Discussion

### 4.1. Analysis of Emission Spectra of Flavins and Whole Bacterial Cells

The peak of the flavin standards is found between 510 and 550 nm, but the three spectra were indistinguishable, meaning that we had to take the autofluorescence of all three as one singular value. The same problem occurred later when we could not differentiate between their average fluorescence lifetimes. Based on this observance, whenever we mention flavins’ fluorescence, the fluorescence of FAD, FMN, and riboflavin combined as one value is meant. As was written in Section 3.1, one team led by Galbán [26] reported the average fluorescence lifetime of riboflavin to be 5.06 ns, FMN to be 4.7 ns, and FAD to be 2.27 ns. The second team, led by Albani, supported these values, and their experiments yielded values of τ_FAD_ = 2.3 ns and τ_FMN_ = 4.7 ns [27]. We strongly agree with these values because of the average fluorescence lifetimes we concluded to be connected with flavins in the cells (see Section 3.3).

As seen in Figure 2, the fluorescence maximum is found at approx. at 515 nm. These spectra resemble that of flavins, and, therefore, flavins are considered to be the predominant source of the cells’ green autofluorescence. When compared, the spectra of the two *Cupriavidus necator* strains differ. That could be because of the presence of PHB granules in *Cupriavidus necator* H16—the inner environment is different, and, thus, the absorption and refraction affect the absorbed and emitted light to act differently. As an example, a study that proved the granules produced by *C. necator* H16 scatter UV light, and that makes them most probably UV protectant [35]. Because of the wide peak area and quite significant tail, it is advisable to use dyes other than green fluorescence dyes when conducting multi-parametrical fluorescence experiments. In other words, as the data indicated, green cellular autofluorescence is relatively strong in intensity and can influence the resulted data very strongly—especially when a dye that has emission in the green region is also introduced into the sample. Fluorescence subtraction is a solution, but it takes time, and the analyses are prolonged and substantially more complicated. Moreover, when preparing the samples, it can bring in a quite significant margin of error.

The cell as a whole contains numerous other fluorophores that influence the green fluorescence of flavins. As seen in the spectra presented, there is found, albeit small, fluorescence signal in the red wavelengths area. Flavins do emit slight fluorescence also in the red channel [36]. This reminds us that autofluorescence is also found in the red channel, so it is very important when using red fluorescent dyes, to potentially use fluorescence subtraction or compensation.

FAD in neutral aqueous solution is known to exist in two conformations: a non-fluorescent stacked conformation in which the isoalloxazine and adenine aromatic rings are in close proximity, and a fluorescent open conformation in which the two aromatic rings are separated from each other [32]. The fluorescence lifetime of the stacked conformation is several picoseconds, and that of the open conformation is 2–3 ns, according to the literature. These two conformations have the same peak wavelengths in absorption and fluorescence spectra, so the only way to distinguish between them is the average fluorescence lifetime [33,34,37]. This is a very important piece of knowledge for us since, as is discussed further in the text (see Section 4.3), this explains the results of further experiments.

### 4.2. Using Green Autofluorescence to Monitor the Viability of Prokaryotic Cell Cultures

When monitoring any culture of microorganisms, viability is one of the key factors. To put it simply, it is important to know how many of the cells in that culture are viable and functioning properly. Many fluorophores are used when answering the question of viability. For example, propidium iodide is one of the most widely used [38,39] ever since Boulos and the team published the methodology [40]; however, there are some limits to this essay, such as viable, but non-culturable and stressed cells that cause the results to be misrepresented [41,42,43,44]. Generally, introducing another molecule to the cell means that the results may not be as precise as desired. The main reasons why introducing another fluorophore can be considered result-changing are as follows: (i) the time it takes to prepare the sample, stain the culture, and then analyze it; and (ii) with every introduction of a foreign molecule to the culture, the balance of the whole system may become unstable and affected. In the monitoring of biotechnological processes, the speed of the sample preparation is very important, especially if the physiological state of the culture is important and if the conditions require adjustment [33].

We identified two excitation maxima when analyzing whole microbial cells of *C. necator* cultures—one in the range of 360–370 nm and another around 440 nm, as can be seen in Figure 3. Very interesting is the fact that the excitation spectra of *C. necator* PHB-4 show higher intensity of excitation than that of *C. necator* H16 when the second maxima—and even third, in the case of the PHB non-synthesizing bacteria—is reached. A redshift, when compared to *C. necator* H16, is noteworthy. This is most probably caused by the structure and content of the cell. Pletnev et al. [45] experimented with far-red fluorescent proteins in different environments and found that bathochromic shifts of the spectra are a natural result of the proteins’ interaction with water and other molecules that caused some structural changes in the protein molecules. Thus, our resulting maxima can also be the result of the flavins’ direct closeness to lipophilic PHB granules in *C. necator* H16, causing the mutant’s spectrum to be redshifted, since the flavins are in a more aquatic environment.

One of the prevailing mysteries we experienced while obtaining these data and pictures was why the *C. necator* PHB-4 cells have certain areas with higher intensity of fluorescence that are alternating with areas where the fluorescence is minimal. Other authors observed non-uniform distribution and compartmentation in prokaryotic cells due to the inner structure of the cells; for example, Spahn et al. observed this effect in *E. coli* [46]. *C. necator* and non-spore-forming *Bacillus* cells are in some ways very similar. In Reference [47], Nicola Manzo et al. showed how a *Bacillus* cell can store a spore inside the mother-cell, and the spore subsequently strongly exhibited autofluorescence. Moreover, although *C. necator* does not form spores, there can be some similar mechanisms of autofluorescence placement between these two cultures caused by the presence of the extraordinary amount of PHB granules in cells of *C. necator*, and this could potentially explain why our cells look as though they do in Figure 5.

### 4.3. Analysis of Average Fluorescence Lifetimes

In Table 1, we can observe further interesting differences between *C. necator* H16 and *C. necator* PHB-4. The ratio of the fluorescence intensity of τ_2_, the shorter average fluorescence lifetime, to intensity of τ_1_, the longer lifetime, is higher in *C. necator* PHB-4. Compare this to Table 2, where we present results from the same type of experiment, but this time, the cells of both strains were stained with BODIPY 493/503 (4,4,-Difluoro-1,3,5,7,8-Pentamethyl-4-Bora-3a,4a-Diaza-Indacene). It is a fluorescent probe that stains lipids. This fluorophore requires relatively short incubation times—15 min (which is important mainly in analyses for which it is crucial to maintain the momentary balance of polymerization and depolymerization); predominantly, there is no need for permeabilization and fixation of the bacterial cells. Moreover, the quantum yield is higher than that of Nile red, which is used for similar purposes. For example, in the screening of soil bacteria for PHB production, the viability of the bacteria correlated to the PHB volume [48]. These characteristics are probably due to lower emission wavelengths—the BODIPY dyes’ emissions are usually in the range of 500–560 nm [49]. BODIPY 493/503 is widely used in analyses of PHB-producing bacteria or in PHB-production monitoring; for example, it was used in *Halomonas boliviensis* [50], *Pseudomonas putida* in carbon limited environment [51], *Bacilus cereus* [52], or when monitoring PHBV copolymer production by *Arxulla adeninivorans* [53]. As was further proven, BODIPY 493/503 has much lower background noise when compared to Nile red, and the repetition of analyses is more reliable [54]. An obstacle when staining with lipophilic probes such as BODIPY is that they are bound to every lipidic molecule, not only PHB granules. It is given that the PHB granules’ fluorescence is much higher because of their large surface in comparison to any other possible site of bonding for lipophilic stains in non-PHB producing cells; this is a basic fact for every PHAs fluorescence analysis there is—flow cytometry and fluorescence microscopy alike.

The reason for this experiment was so that we could see if we could stain the cells with a green fluorescence emitting fluorophore and still observe the average fluorescence times related to autofluorescence. BODIPY 493/503’s average fluorescence lifetime is stated to be 5 ns by the manufacturer, Thermo Fisher. The objective was to see if it is possible to differentiate between the average fluorescence lifetimes of the cell’s flavins and BODIPY 493/503, or if the dye’s intensity is so much higher and our instrument is not able to record the fluorescence lifetime emitted by the flavins contained in the cell.

Exactly two fluorescence average lifetimes of cellular autofluorescence (τ_1_ = 0.8–1.2 ns, τ_2_ = 3.5–4.2 ns) were present in every sample we analyzed, and these can also be used when monitoring bacterial populations. The ratio τ_1_:τ_2_ was roughly 1:1 when analyzing only the cells’ autofluorescence. After adding BODIPY 493/503, the ratio τ_1_:τ_2_ was 1:15. This shows us a clear answer to our questions stated above—BODIPY 493/503’s intensity of fluorescence does mean that we are unable to detect the longer average fluorescence lifetime belonging to the cell’s flavins, even though the concentration of BODIPY 493/503 in the sample was a hundred-times less than is recommended by the manufacturer. However, the good news is that we can use the diluted dye; we can use it fully for its lipophilic staining properties and still be able to detect the autofluorescence of the flavins.

One of our main questions was as follows: “Can we use flavin’s average fluorescence lifetimes as a marker when the cell culture loses viability?” Here we bring proof that the shorter lifetime can still be used in analyses when staining the cells. The fluorescence average lifetime of 0.8–1.2 ns can be used as a stable marker in various biotechnological processes where the assay includes fluorophores with an average fluorescence lifetime longer than 4.2 ns.

### 4.4. Flavin-PHAs Hypothesis

The different τ_1_:τ_2_ ratios in both bacterial strains can be explained by this hypothesis. The average fluorescence lifetimes of τ_1_ = 0.8–1.2 ns and τ_2_ = 3.5–4.2 ns are present in both strains, yet their intensities are different—*C. necator* H16 is about up to ten times brighter in intensity than its mutant *C. necator* PHB-4 (see Section 3.2). PHB granules are universal storage compounds of energy and carbon that are synthesized during times of oversupply with carbon-rich sources. When the bacteria enter the period of starvation, the nutrients for energy and carbon are provided by the depolymerization of the granules. In PHA-accumulating bacteria, PHB synthesis and mobilization occur simultaneously; therefore, PHB metabolism is also termed the PHB cycle [55]. Concentrations of CoA, acetyl-CoA, 3HB-CoA, and NAD^+^/NADH are what determine whether PHB synthesis or depolymerization prevails [56].

PHB synthesis is well understood mainly because PHB is a very important biotechnological product used in medicine, pharmacy, agriculture, etc. Although the strictly isotactic microstructure can be a problem, PHAs can be used for the manufacturing of bottles, containers of various characters, and flexible films; all of them are fully biodegradable [57]. In medicine, one of the uses is in tissue engineering—the hydrophilicity and water uptake of such products can be increased, water contact angles can be decreased, and the possibilities are almost limitless [58].

In *C. necator*, the PHB granules’ surface is covered by six types of proteins: PHB synthases (*pha*C1 and *pha*C2); PHB depolymerases (*pha*Z) [59]; oligomer hydrolases; and other phasins, i.e., *pha*Ps, which cover most of the surface and prevent coalescence of granules, and *phaM*, which binds the granule to nucleotide via *pha*P5 [60].

Under nitrogen- or phosphorous-deficit conditions and an excess of carbon sources, the metabolism of the TCA cycle is repressed, resulting in an increase of acetyl-CoA, which is then redirected to the PHA biosynthesis [61,62,63]. According to Stubbe et al., PHAs synthesizing bacteria possess three-fold the amount of CoA compared to PHA non-synthesizing cells [64]. An increase in ATP and guanosine tetraphosphate levels was observed to be concomitant with intracellular PHA degradation [65].

It is widely accepted that the first step in PHB depolymerization is hydrolysis by intracellular PHB depolymerase, with the product being 3-hydroxybutyric acid (3-HB). Then, there are two possible pathways. In the first one, with the help of dehydrogenase, acetoacetate is produced, which is then activated by 3-ketothiolase to acetoacetyl-CoA, with the final product being acetyl-CoA. It then enters TCA or the glyoxylate cycle. In the second pathway, 3-HB acid is activated by acyl-CoA synthase or thioesterase. Then (R)-3-hydroxybutyryl-CoA is transformed to (S)-3-hydroxybutyryl-CoA, which then enters ß-oxidation. There can be an instance where 3-HB is secreted outside the cell, or, in other words, degraded with secretion into extracellular space [66,67].

Acetyl-CoA is an intermediate of the central carbon metabolism and precursor of the PHB pathway [68]. Catabolism of PHB to acetyl-CoA causes higher autofluorescence intensity in *C. necator* H16 in the sense of higher turn metabolism and the presence of more flavins. A higher concentration of acetyl-CoA also means a higher concentration of succinate dehydrogenase, where FAD <-> FADH_2_. Acetyl-CoA enters the citrate cycle, where FADH_2_ is generated when succinate is oxidized to fumarate. Succinate dehydrogenase is also known as complex II in the respiration chain [66,69].

In ß-oxidation, flavin is present as a part of acyl-CoA dehydrogenase in the step of acyl-CoA being reduced to enoyl-CoA. The link between ß-oxidation and PHB catabolism is only through fatty acids synthesis from acyl-CoA that was obtained by PHB degradation. The obtained FADH_2_ is then transferred to ETF, electron transferring protein, and generated electrons then get through to ubiquinone [70,71].

## 5. Conclusions

Our results indicate that green cellular autofluorescence can be used as a marker for differentiation between live cells, dead cells, and abiotic particles. This green autofluorescence is most probably caused predominantly by flavins, as we demonstrated with the emission spectra of our flavin standards (FMN, FAD, and riboflavin) and the *C. necator* cells, which were very similar.

There are other authors who used this approach when studying flavins’ fluorescence; for example, one very interesting review on FAD in analytical chemistry compared FAD and FMN [26], using FRET to help distinguish between the three most common flavins via catalytic conjugated polyelectrolyte [72]. However, there were also groups that reported very similar results, as we did in this study [21,25].

The intensity of autofluorescence is very different between non-viable and viable cells, but mainly between cells of *C. necator* H16 (capable of PHA production) and *C. necator* PHB-4 (unable of PHA production), with the first-named emitting fluorescence that was almost ten-fold higher. This could be caused by the relationship between PHB catabolism and a higher pool of acetyl-CoA in the PHB-producing cells [64], which could mean a higher amount of flavins. There were also reports of other groups that proved that the higher percentage of PHB the cell produces, the higher the fluorescence intensity is [17,73].

Three average fluorescence lifetimes were found: 3.8–4.2 ns, 0.8–1.2 ns, and one that was several picoseconds long. These could be because of different conformations of flavins. In the section where we recount using the lipid dye BODIPY 493/503, we found that the average fluorescence lifetime, the interval of 0.8–1.2 ns, was always present. As was stated before, Galbán et al. [26] and Albani [27] reported the average fluorescence lifetimes of riboflavin to be 5.06 ns; FMN, 4.7 ns; and FAD, 2.27 or 2.3 ns. This leads us to believe that the longer average fluorescence lifetime is mainly connected to riboflavin and FMN fluorescence emission, with FAD being mostly responsible for the times around 1 ns. This leads us to the belief that this fluorescence lifetime could be used even in staining assays as a marker of the cells’ presence in this combination of autofluorescence and dye.

## Figures and Tables

**Figure 1 microorganisms-10-01179-f001:**
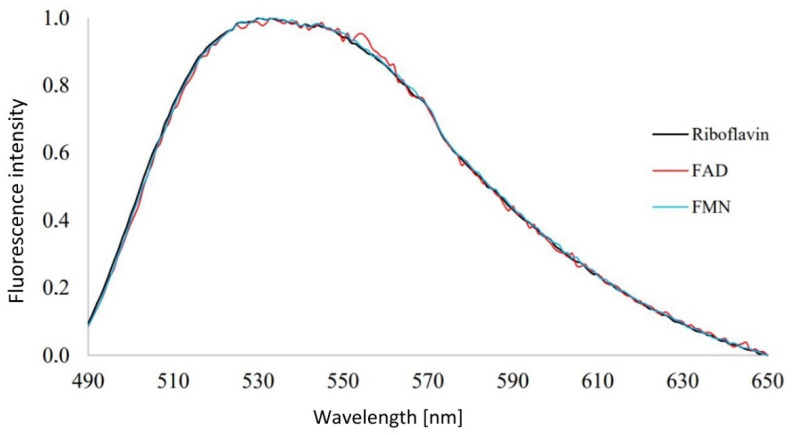
Emission spectra of the flavin standards. Excitation 467 nm, the concentration of the aqueous solution 10^−6^ mmol in 25 mL Erlenmeyer flasks, stored at 4 °C, in the dark, at lab temperature. FMN, flavin mononucleotide; FAD, flavin adenine nucleotide.

**Figure 2 microorganisms-10-01179-f002:**
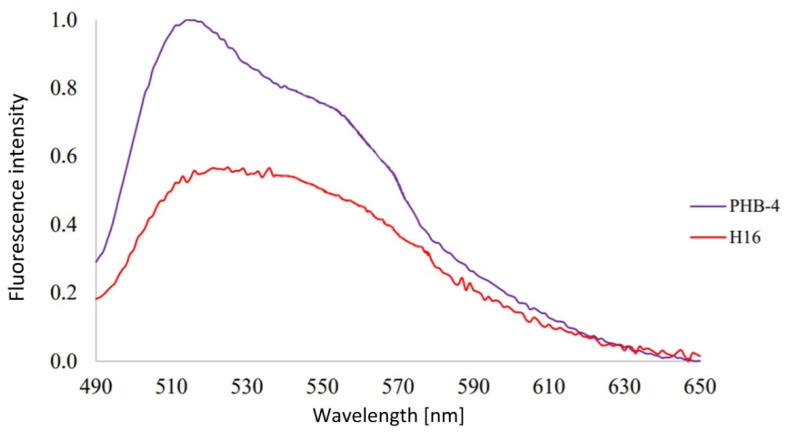
Emission spectra of the two *Cupriavidus necator* cultures used in this study. The concentration of the cells—10^6^ cells in 1 mL PBS (phosphate buffered saline) buffer, 7.4 pH, with 467 nm excitation.

**Figure 3 microorganisms-10-01179-f003:**
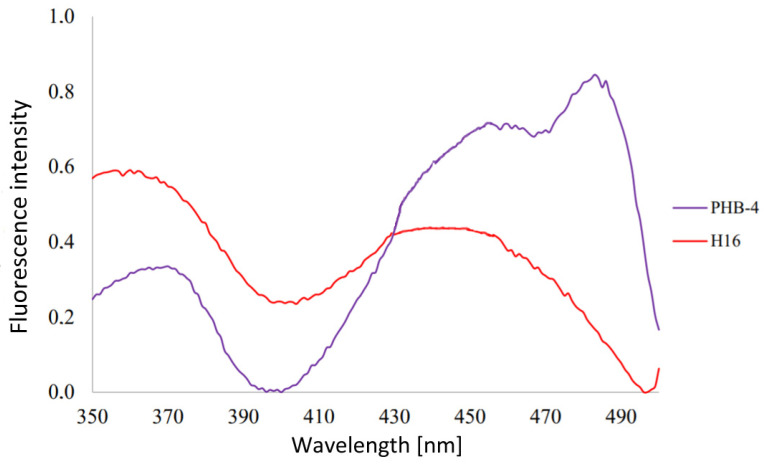
Excitation spectra of *C. necator* cell cultures for emission of 525 nm. The concentration of the cells—10^6^ cells in 1 mL PBS buffer, 7.4 pH.

**Figure 4 microorganisms-10-01179-f004:**
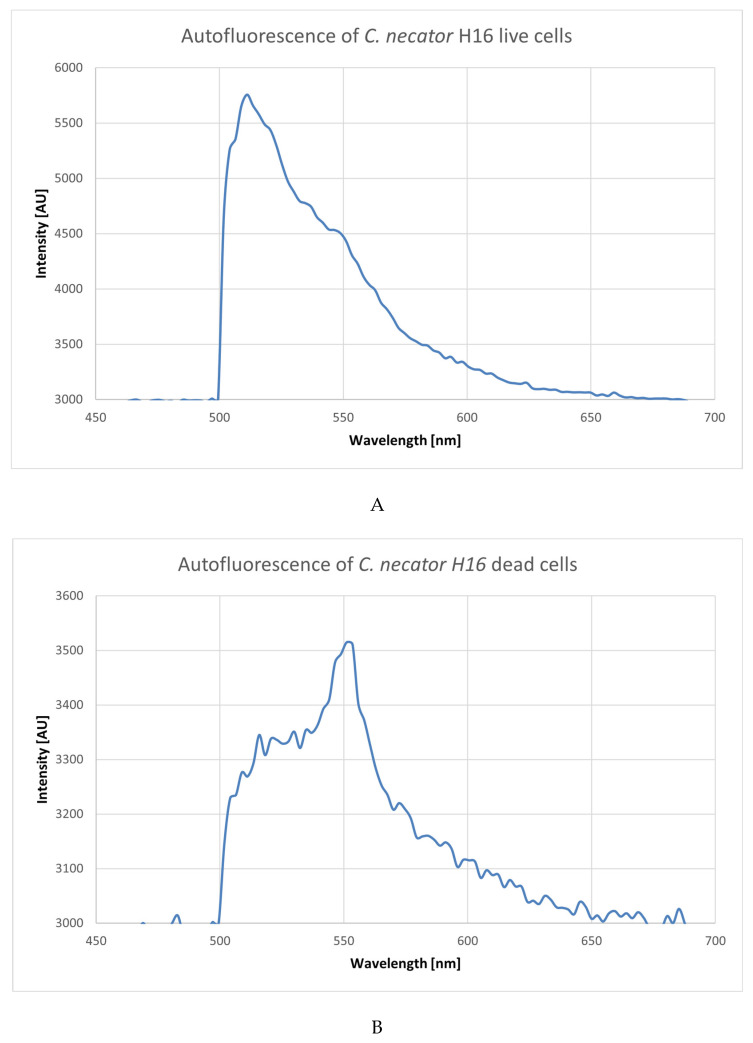
UV excitation with 266 nm laser. Emission spectra of the cultures of *Cupriavidus necator* H16 (**A**,**B**) and *Cupriavidus necator* PHB-4 (**C**,**D**), viable (**A**,**C**) and non-viable (**B**,**D**) cells. The cell concentration was 10^6^ in 1 mL PBS buffer, 7.4 pH. The intensity is the same in all four cases—55 μJ.

**Figure 5 microorganisms-10-01179-f005:**
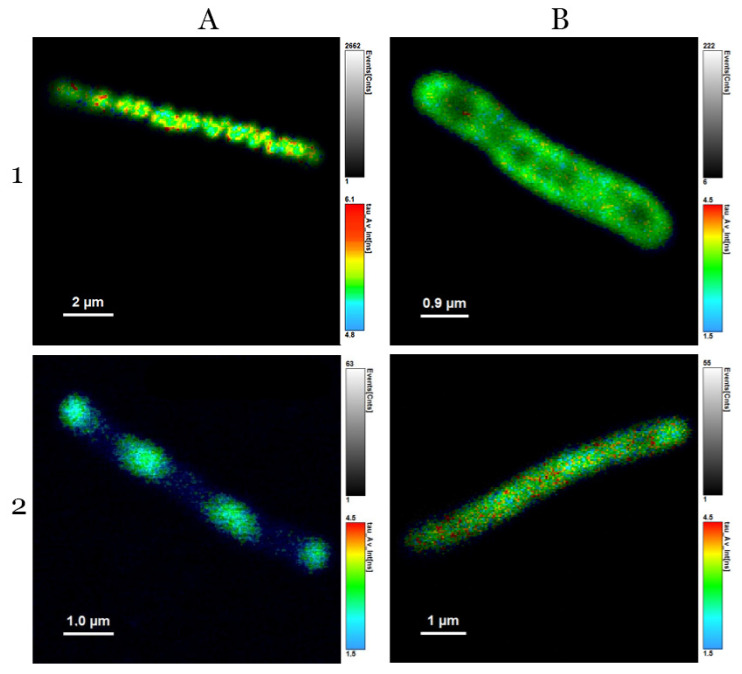
*Cupriavidus necator* H16 (1) and *Cupriavidus necator* PHB-4 (2), live cells (**A**) and dead cells (**B**); excitation with blue laser—467 nm.

**Figure 6 microorganisms-10-01179-f006:**
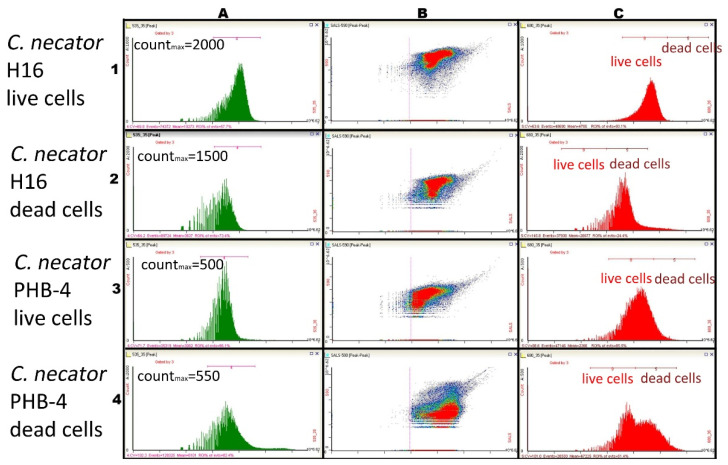
Autofluorescence of cells (**A**); scattergram of the culture (**B**); live/dead cells dyed with propidium iodide (**C**); *Cupriavidus necator* H16 live cells—autofluorescence count maximum 2000 (1); *Cupriavidus necator* PHB-4 live cells—autofluorescence count maximum 1500 (2); *Cupriavidus necator* H16 dead cells—autofluorescence count maximum 500 (3); *Cupriavidus necator* PHB-4 dead cells—autofluorescence count maximum 550 (4).

**Figure 7 microorganisms-10-01179-f007:**
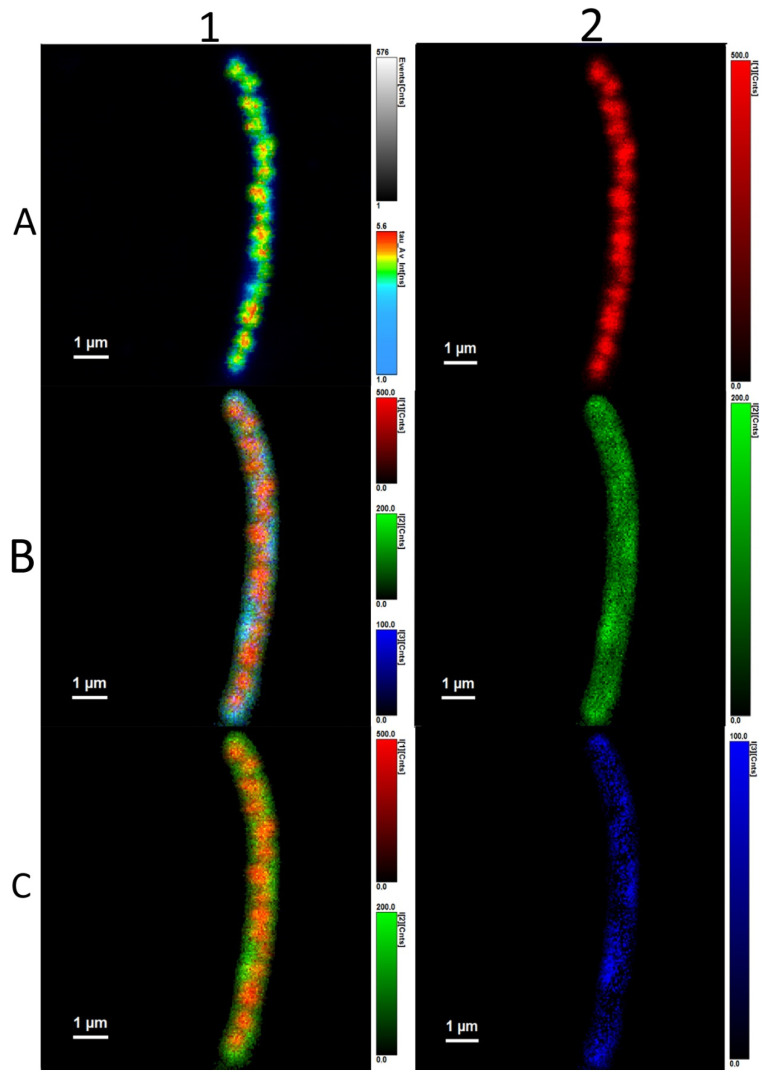
*Cupriavidus necator* H16. Autofluorescence (**1A**), all three different lifetimes of fluorescence of flavins observed are combined in one picture (**1B**), two average fluorescence lifetimes of flavins found (**1C**). The three different fluorescence lifetimes are imagined here separately: 4.800 ± 0.015 ns (**2A**), 1.220 ± 0.044 ns (**2B**), and 0.00532 ± 0.00144 ns (**2C**).

**Figure 8 microorganisms-10-01179-f008:**
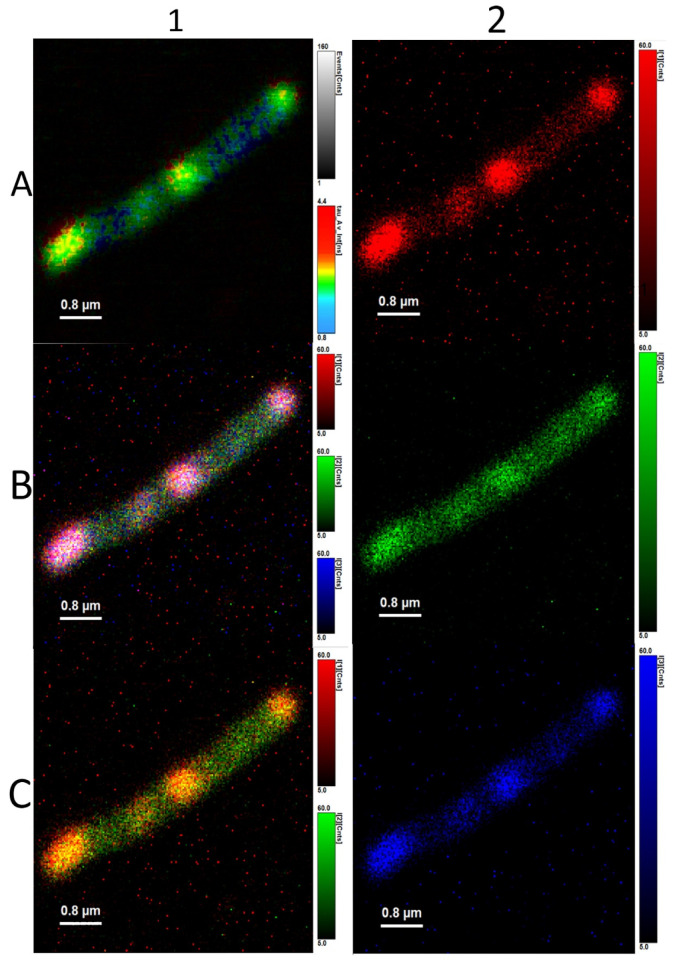
*Cupriavidus necator* PHB-4. Autofluorescence (**1A**), all three lifetimes of fluorescence of flavins observed are combined in one picture (**1B**), two average fluorescence lifetimes of flavins found (**1C**). The three different fluorescence lifetimes are imagined separately: 4.330 ± 0.062 ns (**2A**), 1.180 ± 0.029 ns (**2B**), and 0.00773 ± 0.00062 ns (**2C**).

**Figure 9 microorganisms-10-01179-f009:**
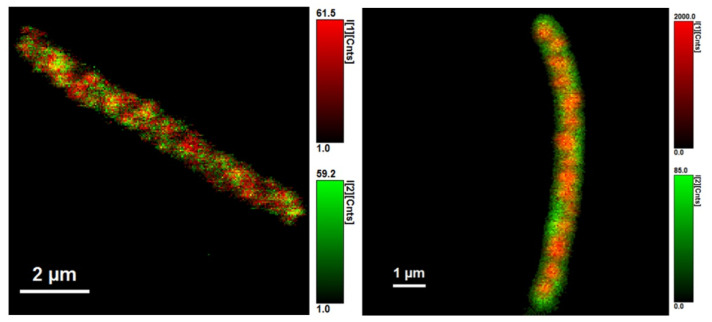
The fluorescence average lifetime of 0.8–1.2 ns can be used as a stable marker (not only of viability) in various biotechnological processes that use fluorophores with average fluorescence lifetime longer than 4.2 ns.

**Figure 10 microorganisms-10-01179-f010:**
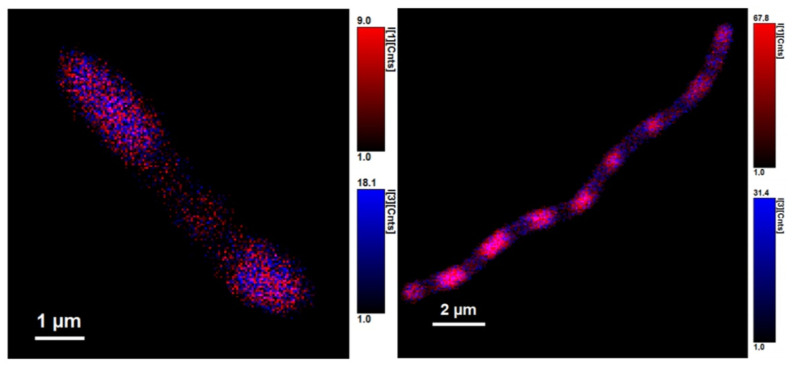
Cupriavidus necator PHB-4.

**Table 1 microorganisms-10-01179-t001:** Fluorescence intensity, A [kCnts]; and average fluorescence lifetimes, *τ*_1_ and *τ*_2_, for chosen samples.

	Sample	A_1_ [kCnts]	A_2_ [kCnts]	*τ*_1_ [ns]	*τ*_2_ [ns]	A_2_/A_1_
*Cupriavidus necator* H16	1	0.105 ± 0.004	0.197 ± 0.007	4.17 ± 0.02	1.29 ± 0.06	1.9
	2	0.452 ± 0.008	0.775 ± 0.023	5.04 ± 0.04	1.13 ± 0.04	1.7
	3	0.262 ± 0.009	0378 ± 0.016	4.54 ± 0.05	1.22 ± 0.03	2.2
	4	0.279 ± 0.015	0.700 ± 0.008	4.60 ± 0.12	122 ± 0.04	2.5
	5	0.320 ± 0.020	0.751 ± 0.012	4.30 ± 0.11	1.15 ± 0.05	2.3
	6	0.034 ± 0.003	0.082 ± 0.004	3.80 ± 0.11	0.92 ± 0.05	2.4
	7	0.061 ± 0.005	0.103 ± 0.003	4.l0 ± 0.13	1.14 ± 0.07	1.7
*Cupriavidus necator* PHB-4	1	0359 ± 0.008	1.270 ± 0.015	4.09 ± 0.04	1.12 ± 0.02	3.5
	2	0.147 ± 0.006	0.479 ± 0.012	3.80 ± 0.06	1.11 ± 0.03	3.3
	3	0.594 ± 0.012	2.420 ± 0.027	3.61 ± 0.05	1.07 ± 0.01	4.1
	4	0.740 ± 0.020	3.210 ± 0.037	3.38 ± 0.03	1.06 ± 0.02	4.3
	5	0.072 ± 0.003	0.220 ± 0.007	3.65 ± 0.07	0.88 ± 0.03	3.1
	6	0.052 ± 0.006	0.161 ± 0.006	4.00 ± 0.22	0.90 ± 0.06	3.1
	7	0.288 ± 0.015	0.979 ± 0.008	4.33 ± 0.06	1.18 ± 0.03	3.4

**Table 2 microorganisms-10-01179-t002:** Fluorescence intensity, A [kCnts]; average fluorescence lifetimes, *τ*_1_ and *τ*_2_, for samples dyes with BODIPY 493/503.

	Sample	A_1_ [kCnts]	A_2_ [kCnts]	*τ*_1_ [ns]	*τ*_2_ [ns]	A_2_/A_1_
*CN* H16	1	6.000 ± 0.018	0.549 ± 0.023	5.06 ± 0.01	1.50 ± 0.15	0.09
	2	5.240 ± 0.022	0.571 ± 0.017	5.11 ± 0.01	1.22 ± 0.10	0.11
	3	6.470 ± 0.028	0.596 ± 0.022	5.19 ± 0.01	1.22 ± 0.09	0.009
*CN* PHB-4	1	0.412 ± 0.019	0.476 ± 0.008	5.08 ± 0.08	1.40 ± 0.11	1.15
	2	0.273 ± 0.005	0.339 ± 0.007	5.07 ± 0.07	1.29 ± 0.03	1.24
	3	0.295 ± 0.016	0.364 ± 0.016	5.10 ± 0.03	1.18 ± 0.08	1.23

## Data Availability

Not applicable.

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
