# Peer review of "Use of Flavin-Related Cellular Autofluorescence to Monitor Processes in Microbial Biotechnology"

_microorganisms, 2022, doi:10.3390/microorganisms10061179_

Round 1

Reviewer 1 Report

After reading the improved version of the Manuscript, I can see a response to the previous comments of the reviewers. In the current form, I recommend publishing the article in Microorganisms.

Reviewer 2 Report

Manuscript ID: microorganisms-1682614

Title: Use of flavin-related cellular autofluorescence to monitor processes in microbial biotechnology

Dear authors, I am impressed with the revision work of the manuscript. Before publishing, please correct some small editorial errors:

Line 362: Figure 11.C. - spacing problem

Line 435: inFigure 3. - spacing problem

Line 512: 4.1. Flavin-PHA hypothesis – numebring problem – please check, 4.4?

Kind regards

Reviewer

This manuscript is a resubmission of an earlier submission. The following is a list of the peer review reports and author responses from that submission.

Round 1

Reviewer 1 Report

Dear authors, 

The use of flow cytometry for quantitative measurement of PHB in Cupriavidus necator was previous reported: Mengxing Li, Mark Wilkins,
Flow cytometry for quantitation of polyhydroxybutyrate production by Cupriavidus necator using alkaline pretreated liquor from corn stover,
Bioresource Technology, Volume 295, 2020, 122254, ISSN 0960-8524,
https://doi.org/10.1016/j.biortech.2019.122254. There are several issues related to the methodology and results - there is no statistical analysis of the obtained data, the results are rather descriptive not quantitative, gating strategies are not included, the graphs referring to the flow cytometry results should include the legend on the axis, there are no marker included for clear discrimination between the dead and live cells. You might consider plotting the average fluorescence lifetimes values. 

My recommendation for the authors is to improve and resubmit the manuscript. 

Thank you very much 

Kind regards

Reviewer 2 Report

pdf file

Reviewer 3 Report

Autors used green autofluorescence induced by flavins as a tool to monitor fermentation employing bacterium Cupriavidus necator. The autofluorescence was used to distinguish microbial cells from abiotic particles in flow cytometry assays and it was also used for the determination of viability or metabolic characteristics of the microbial cells. The results indicated that green cellular autofluorescence can be used as a marker for differentiation between live, dead cells, and abiotic particles. The intensity of autofluorescence was a very different mainly between cells of C. necator - capable of PHA production and - unable of PHA production. The study proved as well that the fluorescence lifetime could be used even in staining assays as a marker of the cells´ presence.The results of these studies are very interesting, promising and  useful. The methods presented are clear, repeatable and possible to implement. Nevertheless, the manuscript needs to be corrected. Changes and suggestions would be found in the attached PDF-file, in the specific comments for all sections of the paper. 
